# Approach for Intuitive and Touchless Interaction in the Operating Room

**Alexandre Hurstel \*** 🆔 **and Dominique Bechmann**

ICube (IGG team), Université de Strasbourg, CNRS (UMR 7357), 67081 Strasbourg, France; bechmann@unistra.fr
\* Correspondence: a.hurstel@unistra.fr

**Abstract:** The consultation of medical images, 2D or 3D, has a crucial role for planned or ongoing surgical operations. During an intervention, this consultation induces a sterility loss for the surgeon due to the fact that the classical interaction devices are non-sterile. A solution to this problem would be to replace conventional devices by touchless interaction technologies, thereby enabling sterile interventions. In this paper, we present the conceptual development of an intuitive "gesture vocabulary" allowing the implementation of an effective touchless interactive system that is well adapted to the specificities of the surgical context. Our methodology and its implementation as well as our results are detailed. The suggested methodology and its implementation were both shown to be a valid approach to integrating this mean of interaction in the operating room.

**Keywords:** touchless interaction; mid-air interaction; gesture; surgery

---

## 1. Introduction

In modern surgery, medical data visualization has taken a crucial role. This visualization most often consists of data representation through 2D or 3D images. These forms of representation are an important aid for the planning of operations (pre-op), as well as their proceedings (per-op) as a consultative support for the surgeons. It is the latter case that has taken our upmost attention in this paper, since the per-op data consultation remains problematic.

Currently, in its most common occurrence in operating rooms, this per-op data consultation is carried out using a classical desktop computer setup: an isolated computer using the classic mouse and keyboard devices as a mean of interaction with the data visualization system. The main and most important problem brought by this setup is that of sterility loss. It is well-known and understood that, during an ongoing surgical operation, sterility must be preserved at all time. However, the use of a mouse and a keyboard do not fit with this necessity since they are not sterile devices and their sterilization is difficult to achieve. This means that the use of the desktop computer setup forces the surgeon (or any other member of the surgical team) to be de-sterilized each and every time they have a consultative need for the current operation. This also means that this person, once the data consultation is done, has to be re-sterilized to proceed with their task in the surgery. As a result, the simple act of data consultation during an ongoing surgical operation becomes a notably time-consuming process which, in this context, can become a critical constraint.

A common bypass for this problem is to use a task delegation strategy: the leading surgeon delegates the consultation task to one of his assistants, therefore preserving himself from the de-sterilization/re-sterilization process. However, while this is an acceptable solution, we consider it to be "not good enough" since this solution can induce an expertise loss on consultation since the assistant may not have the same expert eye as the leading surgeon would. Furthermore, we believe that, if the de-sterilization/re-sterilization constraint were to be lifted, the data consultation task would

become a more "affordable" tool, in term of constraints, and therefore more reliable and help improve the overall workflow of the surgical operation.

Our solution to overcome the sterilization problem of the medical data consultation system is based on a simple idea: there is no sterility loss during the interaction with the system if no touching is involved in the process. Fortunately, touchless interaction devices such as the Microsoft Kinect^TM and the LeapMotion^TM, which were developed and made accessible on the market in the recent years, have made the creation and development of mid-air gesture based touchless interaction systems possible for any application without heavy technical constraints. Therefore, the conception of a medical data consultation system free of any physical interaction and thus free of the sterility constraints found in operating rooms is made more affordable and accessible than ever before.

The conception of such system is obviously not trivial. Many aspects of this system have to be thought through and studied, namely usability considerations in order to develop with a solution that is truly usable and effective. In Section 2, we review our approaches regarding the development of a gesture based touchless interaction adapted to the specific context of use that is the operating room.

*Presented Work*

This paper first presents, in Section 2, previous related studies that have explored and outlined the challenges raised by the design of intuitive touchless interaction and how these challenges have been approached in the context of the surgical environment. Section 3 presents the methodological approach we have adopted, the ensuing criteria our work revolves around, the needs we aimed to cover, and the technical choices we made to cover this issue. Sections 4 and 5, respectively, present the methodology and its implementation into experimental protocols. Finally, Section 6 presents the provisional results we obtained and whether they validate our approach for designing intuitive and touchless interactions system for the specific surgical context.

## 2. Related Works

### 2.1. General Touchless Interaction

Even if devices such as the Microsoft Kinect^TM and the LeapMotion^TM have made touchless interactions system accessible to the general public, gesture-based interactions have been a recurrent subject for research in the field of VR (Virtual Reality) for many years. Bobillier-Chaumon et al. [1] studied standardized interactions and how common cultural references have made these interactions accessible, easy to implement, and reusable across various applications and cases of use. However, Norman and Donald [2] showed the limit of standardized interactive gesture based systems by pointing out the fact that gestural standards cannot take into account inter-individual or cultural differences (e.g., hand waving means "hello" in western culture but means a rejection or disapproval for Indian users). Research studies such as those of Jego [3] have strongly suggested that standardized and pre-defined gesture vocabularies for a cross application use are not optimal since gestures and their meaning are context sensitive. What we refer to, here and in the rest of this paper, as a gesture vocabulary is a set of gestures to which the available actions of the system are associated. A gesture vocabulary is therefore the list of the gestures a user has to induce to produce the associated action in a specific context (e.g., inducing finger drag on a mobile application to produce a scroll). Our work does not aim at providing a universal interaction system, but a system that is operational in a specific context and environment. This entails that we do not need adaptive or custom user-defined gestures vocabularies for our touchless interaction system.

### 2.2. Touchless Interaction within the Surgery Context

In recent years, due to the regain in popularity of the touchless interaction devices, research in gesture-based interaction has integrated various domains of application, including the medical data consultation scenario.

Graetzel et al. [4] gave an overview of the benefits of touchless interaction systems for medical data consultation by re-mapping a mouse cursor to a system using computer vision to track users hands. While overcoming the problem that we identified (sterility maintenance), this system was not easy to use, was not effective, had a low usability, and induced fatigue for the users. This is due to the lack of a specific gesture design, and also to the fact that an effective function mapping cannot be provided by a simple mouse substitution. A further step was taken by Tan et al. [5] who integrated the interactive features offered by the Microsoft Kinect^TM over the DiCOM software TRICS. Bizzotto et al. [6] took a similar approach using the LeapMotion^TM over the OsiriX software. Both studies mapped directly their respective DiCOM software's consultative functionalities with the recognized set of gestures by the device. Both showed encouraging results as these setups were tested during live operations and received positive feedback and demonstrated that touchless interaction in surgery is met with enthusiastic expectations by the surgeons. However, again, the simple mapping of functionality to gesture lack in usability that could perhaps have been offered if the interactive system had been developed using a specifically designed gesture vocabulary paired with refined functionality adapted and designed for gesture control. Furthermore, for example, Silva et al. [7] elaborated an interactive system specifically designed for the consultation of medical images. Although this work has shown positive and promising results and provided a solid user evaluation, the system still lacks in gesture intuitiveness.

As the interest for touchless interaction for surgery grew, the need for clear methodology for designing such interfaces became more evident. O'Hara et al. [8] made a significant contribution regarding this topic by outlining the issues and challenges posed by such interaction design in this context. Firstly, one of these concerns is the notion of "expressive richness", the ability to map a large set of gestures with a large set of functionalities. Secondly, the question of the use of one- or two-handed gestures and their impact on the usability of the interface arises from this study. Thirdly, the concerns over "mundane gestures" (gestures produced without the intention of interaction) and the interferences between gesture transitions are also brought to light. Finally the issue of engagement and disengagement gestures is also called into question, as a means of avoid unintentional gestures outside of any interacting phase, and as a way of keeping track of the active user in a multi-user environment. To study and address these challenges, O'Hara et al. designed a system using a Microsoft Kinect^TM captor, exploiting its detection capabilities to ensure the expressive richness of said system. This implemented interactive system relied on bi-manual or mono-manual gestures, depending on the interaction. It also exploits voice command as a mean to address the engagement/disengagement issue and uses a time-based lock feature in which holding a hand posture for a fixed amount of time is necessary to activate a specific functionality. From this work, we remark that: firstly, most of the concerns are, at least partially, dependent on the captor; and, secondly, the set of gestures controlling this system, even though based on end-user interviews, are seemingly chosen directly by the developers.

## 3. The Search for the Optimal Interactive System

### 3.1. The Search for a Gesture Vocabulary

#### 3.1.1. Methodology

As pointed out early on by Nielsen et al. [9], the development of a gesture-based interactive system is neither trivial nor can be overlooked as a side matter. The authors pointed out, and many others subsequently, for example Jego [3], that the gesture vocabulary (i.e., the set of gestures to which are associated the available actions of the system) cannot be regarded as universal and is, in fact, context specific.

The gesture vocabulary, to be feasible, needs to be positively evaluated on certain criteria (Keates et al. [10]) that are, most importantly, intuitiveness, ease of use (comfort), and ease of remembrance. Following the advice given by Nielsen et al. [9], we ruled out a technology-based

approach to develop our vocabulary since, by essence, this method does not aim to optimize the end-user's comfort. The reason for which we emphasize intuitiveness in this work is the belief that intuitiveness and ease of use should not be regarded as a secondary goal for any interactive systems or technology, even experimental ones, but should be regarded as of the utmost priority. To this end, methods, such as User Centered Design methods that revolve around advanced user acknowledgement and involvement during the design process, have proven to be effective. The latest evolution among these methods is known as the User Experience (UX) approach. However, as Law et al. [11] explained, such methods are not yet entirely suitable for research due to the fact that they partially rely on qualitative measurements and do not have the luxury to rely on scientific consensus. Furthermore, for example, Lallemand and Gronier [12] explained that these methodologies are best used in industrial contexts given a wide range of participating users and a considerable time allocation dedicated to their implementation. Nonetheless, even if not entirely applicable, these methods give us precious guidelines onto what aspects should be closely taken into consideration in order to develop and refine our gesture vocabularies.

### 3.1.2. Our Criteria

Following these guidelines, we propose the following criteria to evaluate gestures according to Keates et al. [10]:

**Ease of use**: Can the gesture allow for task completion while minimizing the user movements?
**Stress**: Does the gesture requires strain inducing stressful or uncomfortable postures?
**Performance**: Can a gesture be used quickly for a specific task in terms of completion time, and with a reduced number of trials?
**Shared intuition**: Are similar gestures chosen by different users? If so, is it to generate the same interaction?
**Coherence**: Is the same gesture consistently chosen for the same interaction by the same user?
**Memorization**: Is a user able to remember a chosen gesture for a specific interaction?

These criteria, applied to the gestures to collect from users, allow us to obtain a gesture vocabulary built around a refined set of gestures that would be assessed for relevance and intuitiveness.

### 3.2. Identifying the Needs

### 3.2.1. Cases of Use

The development interactive systems begins by identifying the needs this system has to fulfill. Since the end-goal of our work is to replace existing interactions, we start by studying existing cases of use that fulfills these needs. In our case of medical images consultation, we focused on cases of use regarding two types of images:

- 2D tomographic images consultation that, basically, consists of a stack of images that shows organ sections obtained by radio or CT-scans. This case allows interacting with each image individually and navigating within this image stack.
- 3D images consultation that are virtual reconstruction of an organ or group of organs gathered in one structure. This case allows interacting with one 3D model at a time as well as with the whole set of models.

### 3.2.2. Extraction of Targeted Interactions

Regarding the 2D consultation case of use, the programs currently used by surgeons are DiCOM software such as OsiriX. These programs are advanced and offer a very wide range of features, and therefore in interactions. To isolate the interactions relevant to our case of use, an end-user, a fellow-surgeon accustomed to DiCOM software usage for tomographic images consultation in per-op,

gave us a demonstration of the features and interactions in use for this case. We were also allowed to visit an operating room, during an ongoing operation, to have a better view of the layout of this specific workspace while in use. It appeared that the specific set of files (i.e., tomographic scans) corresponding to the patient data was composed and selected in pre-op, meaning that the relevant interactions, for the surgeons, are the interactions induced on the pre-selected scans. Therefore, interactions such as file selection or software settings are not to be considered as target interactions since they are not subject to the initial sterility loss problem. If there is a need for the use of such features, they are uncommon during the operation and the necessary interactions are done by operators who works behind a sterile curtain and are not directly involved in the ongoing sterile operation.

The 3D consultation case of use was provided to us by analyzing the "VP Planning" software. This software is distributed by "Visible Patient", a partner company within the research project that funds our work, and is effectively used by the surgeons and fellow-surgeons we interviewed. This software is available for personal computers or as an application for tablets. For our work, we analyzed the latter version, for simplicity since it is reduced to essential consultation interactions, as explained and confirmed by the end-users. Thus, for this case, extracting the relevant interaction is equivalent to extracting the few interactions implemented by this application.

### 3.2.3. Targeted Interactions

Once our reference cases of use were identified and analyzed, we extracted "interaction taxonomies". These taxonomies allowed us to isolate all of the functionalities that our gesture vocabulary has to target. The task was then to associate relevant and meaningful gestures with each identified functionalities from the taxonomy.

For both of our studied cases of use:

**Translation**: The user should be able to induce a translation on the displayed image. Concerning the 3D case of use, this translation is constrained to the plane containing center of the object and parallel to the screen.
**Zoom**: The user should be able to enlarge or shrink his view of the image or 3D-model.

Furthermore, each case of use possesses its own interaction types. Regarding the consultation of tomographic images (2D case), the specific interactions are:

**Stack browsing**: The user should be able to browse through all the stacked sections of organs that make up this case of use. The user should have control over the browsing speed, being able to browse through a large set of image quickly while still being able to isolate and stop over a specific image.
**Contrast and brightness management**: The user should be able to increase or decrease both the contrast and the brightness of the image.

Regarding the 3D case of use, the specific interactions are:

**Rotation**: The user should be able to induce global rotations over the structure containing the 3D-models of organs. In "VP Planning", our case of use, two types of rotations are available: A 2 degrees of freedom rotation around the X and Y axes, and a rotation constrained around the Z-axis.
**Hide/show individual 3D-models**: The user should be able switch, "on" or "off", the display of selected organ 3D-models.

### 3.3. Working with LeapMotion$^{TM}$

For our work, we chose to work using the LeapMotion$^{TM}$ (LM) device. This is a USB device using monochromatic cameras and infrared LEDs for hand tracking purposes. Its relatively small size (8 cm × 1.5 cm × 3 cm) however constrains it to a reduced detection area (~1 m hemispherical area).

The targeted environment was central in the choice of using the LM. Firstly, we focuses exclusively on hand gestures since a surgeon, during an operation, is considerably limited in his movements. Indeed, the positions of the arms and hand are restricted by a virtual sterility window: the hands have to be maintained below the shoulders and over the hips. Secondly, the small size of the LM combined with its restricted detection area means that its integration into a work environment has a minimal impact while not being too cumbersome.

It is also worth noting that the use of this captor also minimizes some of the concerns brought to light earlier by O'Hara et al. [8] such as the interference caused by nearby members of the operating team, due to the mentioned small dedicated space required. This aspect also partially addresses the engagement/disengagement concerns since getting out of the detection zone is simple and works fine for such feature. Finally, the use of the LM allowed focusing on hand gestures, thus designing gesture vocabularies that potentially rely solely on hand postures, which offers an already decent range of "expressive richness" with minimal movements or repositioning from the user.

Throughout the rest of this paper, "captor" or "device" mainly refer to the LM. Thus, our experimental systems are to be considered as making use of this device.

## 4. Approach

### 4.1. User Suggestions for an Intuitive Gesture Vocabulary

The key idea is to observe the gestures that users produces instinctively when confronted with the task of having to produce a given interaction, knowing its expected effects. In other words, for a given set of tasks, where a user has to execute a specific set of well identified interactions, we asked him/her to produce the gesture that he/she deems instinctively the most appropriate or effective. By doing so, every subject that went through such a process provided us with their own gesture vocabulary constructed around their intuitions on how to complete given tasks using mid-air hand gestures. Obviously, such deduced vocabulary should not be regarded as universal and is expected to notably vary from one subject to another. However, a given number of such suggested vocabularies allowed us to study them concerning several aspects. For instance, we could look for recurring types of gestures and observed if those gesture were used often in similar or close fashions. If so, we could use the collected gestures to deduce a general but intuitive gesture vocabulary specific to the tested application.

This type of methodology has been gaining traction in various fields of interaction research in the recent years. Notably, Vogiatzidakis et al. [13] provided a survey over "methods of gestures elicitation" regarding touchless interaction across various domains of application (e.g., virtual reality, desktop, public display, etc.). It appears that the "operating room" was among the least represented with only one study over 47 selected papers, showing the partial lack of such method applications in a domain where the usefulness of touchless interaction and its potential have been demonstrated. Judging from its date of publication, the aforementioned reviewed study, by Jurewicz et al. [14], was designed and conducted in a timetable close to our own. We nonetheless discuss succinctly the inherent differences with our work, despite numerous similarities, further in this paper (Section 6.4).

To collect gestures proposed by subjects according to specific tasks and actions to fulfill, our approach was to propose an experiment based on "Wizard of Oz" (WOz)-typed experiments. These types of experiments consist, as explained by Dahlback et al. [15], in "studies where subjects are told that they are interacting with a computer system through a natural-language interface (author's note: using a given gesture vocabulary), though in fact they are not. Instead, the interaction is mediated by a human operator, the 'wizard', with the consequence that the subject can be given more freedom of expression, or be constrained in more systematic ways, than is the case for existing Natural Language Interfaces". Of course, the interactions are not induced by the subject but are instead induced by the experimenter, unknowingly to the subject. This principle allows the experimenter to observe and collect interactions, in our case gestures, that naturally come up to the user's mind for inducing specific interactions while being confronted to a case of use that mimics closely the targeted case of use.

This method provided as many different vocabularies as there are subjects. Given this, the collected gestures were analyzed afterwards to identify which gestures were the most frequently used by all. It was not expected that strictly similar gestures would be found across subjects, but the idea was to extract gesture components that occur frequently (hand posture, hand displacements, gestural metaphors, etc.).

*4.2. Two-Stage Approach: User Vocabulary Refinement*

The interactions we focused on, as listed in Section 3.2.3, are either familiar to a wide range of general users (translation and zoom), or at least simple to understand in concept (navigating in an image stack, hiding a pointed organ). In addition, our end-users, the surgeons, is a population that has limited availability. Based on these two observations, we decided to divide our approach into a two-stage process.

1. **General users intuition:** The first stage of our approach consisted in extracting gesture intuitions, by the means presented in Section 4.1, from "all-coming" users, who are not necessarily final or completely aware users. The interactions studied were simple and fairly close to interactions that are found in more mainstream imaging and visualization software and, since it would seem that, after end-user consultation (interviews), surgeons appear to have a user profile that does not differ in a significant manner from those of more general users, this led us to consider the feedback of these users as being relevant. The goal of this first stage was to obtain a generic non-specialized intuition gesture vocabulary.
2. **Target users refinement:** The second stage of our approach consisted in, firstly, enriching and completing the previously extracted vocabulary with gesture intuitions for target users, the surgeons, using again a WOz type experiment; and secondly, having these target users evaluate the previous generic vocabulary and using their feedback to rule out or correct gestures. The goal was to refine the vocabulary with expert view.

**5. Protocol**

We used the following protocol to implement this approach.

*5.1. "All-Coming" Subjects Experiment*

5.1.1. Proceedings

For the first stage of our approach, we adopted the following proceedings to apply to the "all-coming" subjects.

1. **Pre-questionnaire:** The subject was asked to create a user profile via a web-form.
2. **Presentation and explanations:** We started off by presenting briefly the motivation of our work to the subject. We then proceeded by presenting the used captor (LM device) characteristics and capabilities.
3. **Interactions presentation:** We then presented all the interactions (listed in Section 3.2.3), for which the user would have to give an intuition. Of course, no interaction was actually induced but merely the consequence on-screen of the interaction. For instance, when we listed the translation interaction, a pre-recorded translating image was shown. The reason for this was that we wanted to ensure that the subject understood the asked interaction and its expected consequence before proposing a gesture intuition for that interaction.
4. **Interaction proposal:** The user was presented with an artificial case of use (see Section 5.1.2) for which he was instructed to propose a gesture as if he were to complete sequentially a specific set of tasks each time they explicitly used a certain interaction. As in accordance with the principle of the WOz experiment, each interaction was actually induced by the operator, while the subject proposed a gesture.

5. **Interaction questionnaires:** The subject was asked to fill a qualitative web-form asking to describe the interaction he/she proposed for each interaction and his/her inspiration. This allowed us to better understand the intention behind the gesture, to detect any eventual discrepancy between intention and realization, and to detect any widely negative memorability factor of the gesture-interaction association.

6. **Evaluation of a pre-defined gesture vocabulary:** The subject was confronted with a pre-defined gesture vocabulary (see Section 5.1.3). The user was asked to reproduce the same interaction as before but this time actually using a functional touchless gesture interface that implements this pre-defined vocabulary that is focused not on intuitiveness but that aims to minimize hand movement and therefore limits fatigue.

7. **Comparison questionnaire:** The subject was then asked to fill in a form that, for each interaction, required him/her to compare his/her intuition given previously, with the interaction imposed by this vocabulary. This was done to implicitly suggest the subject to correct his/her intuition, if the subject deemed it appropriate, considering factors such as minimized motor realization, mid to long term usage, posture comfort, or fatigue. The goal for us was to collect the comparison feedback to observe whether the subject corrected his/her intuition considering those factors without influencing the initial intuition.

### 5.1.2. Artificial Case of Use

Since the "all-coming" subjects were not target users, and therefore do not have experience regarding medical image data consultation, we estimated that it would be confusing and therefore counter-productive to conduct the protocol with the same data as in a target case of use. This is why we chose to conduct this experiment with an artificial case of use that does not require particular knowledge to understand and on which we could ask of the user to perform the same type of interactions to achieve similar goals.

In practice, for the 2D case of use, we substituted the tomographic scan stack by a stack of relatively generic images on which to perform the interactions (zoom, translations, etc.). In this case, to mimic transition through the stack of images, our example was actually made of a few key images (or stable images) and the in-between images in this stack were actually a morphing transition point between two key images, which allowed us to simulate a continuous navigation in a stack of image in a meaningful manner. The goals for surgeons, when consulting a tomographic scan, is to navigate to find and isolate specific image areas; we believe that our artificial case of use provided a suitable equivalence for our protocol.

### 5.1.3. Pre-Defined Gesture Vocabulary

The motivation for a pre-defined gesture vocabulary despite already collecting gestures intuition was their efficiency is not guaranteed. Even if it is easy for subjects to come up with intuitive gestures "on the go", there is nothing to say they are met with efficiency or long-term usability regarding practical qualities that are, for instance, long-term efficiency, avoiding muscular stress within a lasting or repeated usage, reduced cognitive load, or even technical compatibility with the gesture tracking device.

Our pre-defined gesture vocabulary relies on the principle of "triggering gestures". In practice, this means that our vocabulary relies on certain hand postures that trigger our application into associated interactive modes that are, for instance, regarding our 2D case of use: translation mode, zooming mode, stack browsing mode, etc. The chosen mode remains active as long as the subject maintains their hands in the corresponding posture. The interaction is induced through a virtual vector $\overrightarrow{AB}$ computed from $A$, the position to which the posture has been adopted, to $B$, the current position of the hand (see also Figure 1).

All the interactions rely on the same principle: triggering an interactive mode by adopting the corresponding hand posture and then managing the associated interaction continuously with the

displacement from the posture adoption point. This principle shows three advantages: minimized motor realization, reduced muscular stress, reduced cognitive load.

Since our pre-defined gesture vocabulary is single handed, we evaluated it for both the dominant and non-dominant hands. We always evaluated first our vocabulary with the dominant and then the non-dominant hand to observe if the use of the non-dominant hand induced a critical drawback in terms of efficiency. To compare directly the efficiency impact, both evaluations were performed on the same sequence of interactions.

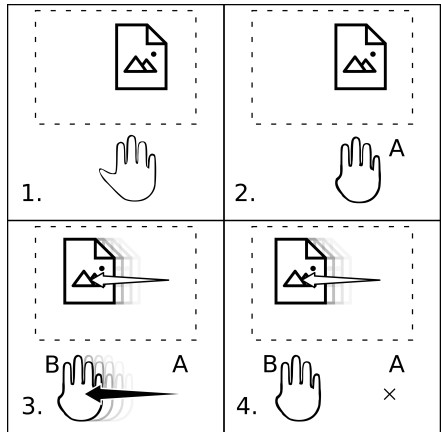

**Figure 1.** Predefined "triggering posture" vocabulary for our 2D case of use. 1. Neutral posture: no interaction; 2. Posture associated to translation mode: translation triggering. Point *A* to which the posture is adopted; 3. Movement of the hand maintaining the posture: virtual vector $\overrightarrow{AB}$, *B* being the current hand position, induces a continuous translation of the image according to $\overrightarrow{AB}$; 4. The hand posture is maintained. No hand movement. The translation according to $\overrightarrow{AB}$ continues as long as the hand posture is maintained.

### 5.1.4. Setup

For our experiment we installed the following setup (see Figure 2) :

- A PC running a custom application (ICube forge (GitLab): https://icube-forge.unistra.fr/hurstel/touchlesssurg_proto1.git) that implemented all interactions and visualization necessary for our case of use. This implementation was written in C++ and uses the Qt library using a QML (Qt Modeling Language) interface to implement the GUI (Graphical User Interface). This program also allowed "playing" pre-configured interaction sequences on command as well as playing, skipping, or reseting these interactions.
- Two monitors. One monitor was used to display the images and the consequences of the interactions to the subject. The other one was for the operator to induce to control the pre-recorded interaction while the subject proposed his/her gesture intuitions.
- A LeapMotion$^{\text{TM}}$ installed on a desk in front of which the subject had to perform his interaction intuition. Although, as explained, the gestures performed did not induce any interaction, the captor was still used to record gesture data as formatted by default by the LM's API.
- A webcam placed between the subject screen and the subject used to film the user gesture, for a post-protocol human recognition and interpretation of the gestures.

We used a custom application instead of existing DiCOM software for our protocol for the sake of flexibility: touchless interaction being an unsupported interactive mode by such software, we wanted to be able to have full control of the display and visual cues. For the 2D case of use, we used a small landmark displayed over the shown image so that the user can keep track of the image displacement during interaction such as a translation and have a better expectation regarding the results of interactions such as zooming, since this land mark helps target specific areas of the

application. In addition, we deemed necessary to add visual cues to help the user locate themselves within the image stack during the stack navigation.

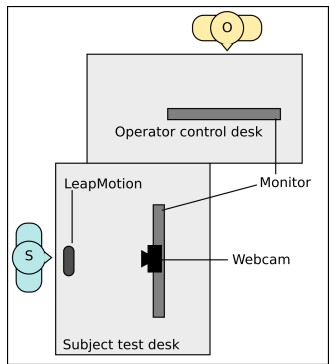

**Figure 2.** Layout of our experimental setup.

*5.2. End-User Subjects Experiment*

For the second stage of our approach, which consisted in refining the previously deduced gesture vocabulary using the expert subjects input, the proceedings were essentially the same. The two major differences within this stage Were:

- The case of use was not artificial, but consisted in regular manipulation and browsing through a stack of CT-scans.
- The pre-defined vocabulary evaluation was replaced by the evaluation of the previously extracted intuitive gesture vocabulary. As before, the evaluation was made two times: first using the dominant hand and then using the non-dominant hand on the same case of use.

For this second stage, the goals were (1) to observe if an intuitive gesture consensus emerges among these expert subjects and, if so, observe and evaluate its degree of difference from the intuitive gestures extracted from "all-coming" subjects; and (2) to evaluate this previous vocabulary and isolate the aspects that need to be refined to perfect its use for the target expert users.

## 6. Provisional Results

*6.1. "All-Coming" Subjects*

We briefly present the observations extracted of our method from the first studied case of use: the 2D tomographic scans consultation. Our experimental "Wizard of Oz" type protocol was carried out on a group of 12 subjects. These observations led to establishing the gesture vocabulary summarized by Table 1.

**Table 1.** First stage extracted vocabulary.

| Interaction | Gesture |
|---|---|
| Translation | • flat open palm; based on movement on XY (desk plane)<br>• grab gesture; displacement on XY |
| Zoom | • triggered by thumb + index extended posture; movement on Y (toward or away from the screen) induces zoom or de-zoom<br>• triggered by thumb + index extended posture; increasing or decreasing distance between the 2 fingers for zoom or de-zoom |
| Stack Navigation | • thumb extended posture; movement on Z (up or down) induces navigation through the image stack |
| Brightness Modification | • thumb extended posture; movement on Y induces increase or decrease of contrast |
| Contrast Modification | • index extended posture; movement on X induces increase or decrease in brightness |

### 6.1.1. Translation

Most of the gestures were performed using a single dominant hand; gestures were always performed within the plane parallel to the desk; the most popular posture was an open hand (42%) followed by a grabbing metaphor (25%).

### 6.1.2. Zoom

Most of the gestures were performed using a single dominant hand (17% of bi-manual); the most popular gesture was a repeated pinch type gesture (58%) seconded by a push-pull typed movement with varying postures (25%).

### 6.1.3. Stack Navigation

Gestures were always performed using a single dominant hand; the most common modality was to adopt a posture (varying) accompanied by an up or down movement (25%); the remaining gestures are varying in modality and nature (pivoting hand gestures, push-pull type mimics, thumbs up/thumb down mimic).

### 6.1.4. Brightness/Contrast Management

The recurring tendency (84%) was to regroup these two interactions under a certain identical modality (posture (25%) and movement direction (42%)), and to discriminate which one is induced with another modality (which hand is used, hand movement, and movement direction). This is likely due to the close nature of these two interactions.

To extract the gesture vocabulary (Table 1), we had to avoid similar or too close gestures for different interactions to minimize the risk of unwanted interaction due to overlapping gestures or missed transitions between them. In addition, we deemed necessary to discard all non-continuous movement-based gestures (i.e., repeated pinching for a continuous zoom), and adapted them to be used by in continuous movement, similarly to our proposed pre-defined vocabulary; this initiative being backed up by user feed back from the comparison questionnaire

### 6.2. End-User Testing

As stated above, we also conducted a "Wizard of Oz" protocol on end-users, who are fellow-surgeons with previous DiCOM software experience. Our observations on the results of this experiment follow those we collected for "all-coming" subjects. When comparing both, no significant differences appeared, in the sense that no new gestures, postures, or interaction metaphors were proposed.

According to the methodology we presented (Section 5.2), we evaluated the performances of the end-users using the gesture vocabulary we were able to deduce from the "all-coming" users protocol. These observations were conducted on a group of five fellow-surgeons.

Each subject's case of use was composed of each type of interactions in a different order. For each type of interaction, three tasks were performed. Additionally, each case of use was done twice for each subject (dominant hand and then non-dominant hand). A general tendency we observed is that the performance increased for each individual task across the two repeated cases of use. The time of completion was reduced as was the number of "gaps", even thoguh the non-dominant hand was used in the second sequence, as shown in Figure 3, indicating that the performance increases due to the learning effect.

Here, what we refer to as the number of "gaps", which is shown in Figure 3a, is the number of times the subject had to correct his current hand position to refine the interaction inducing vector, necessary to attain the goal of the given task. Furthermore we note that the number of gaps decreased for all the interaction of the same type, comfirming our observation. To evaluate the performances we measured, Figure 3b shows the time of completion for each type of task (including both the

dominant and non-dominants hand interactions) for each user. Even though the tasks (of the same type) may differ in terms of amplitude (i.e., the distance between the starting state and the objective), we deemed this to be covered by the fact that our vocabulary allows us to quickly handle both small and large amplitudes of interaction by modulating the virtual vector used to produce the interaction (see Section 5.1.3).

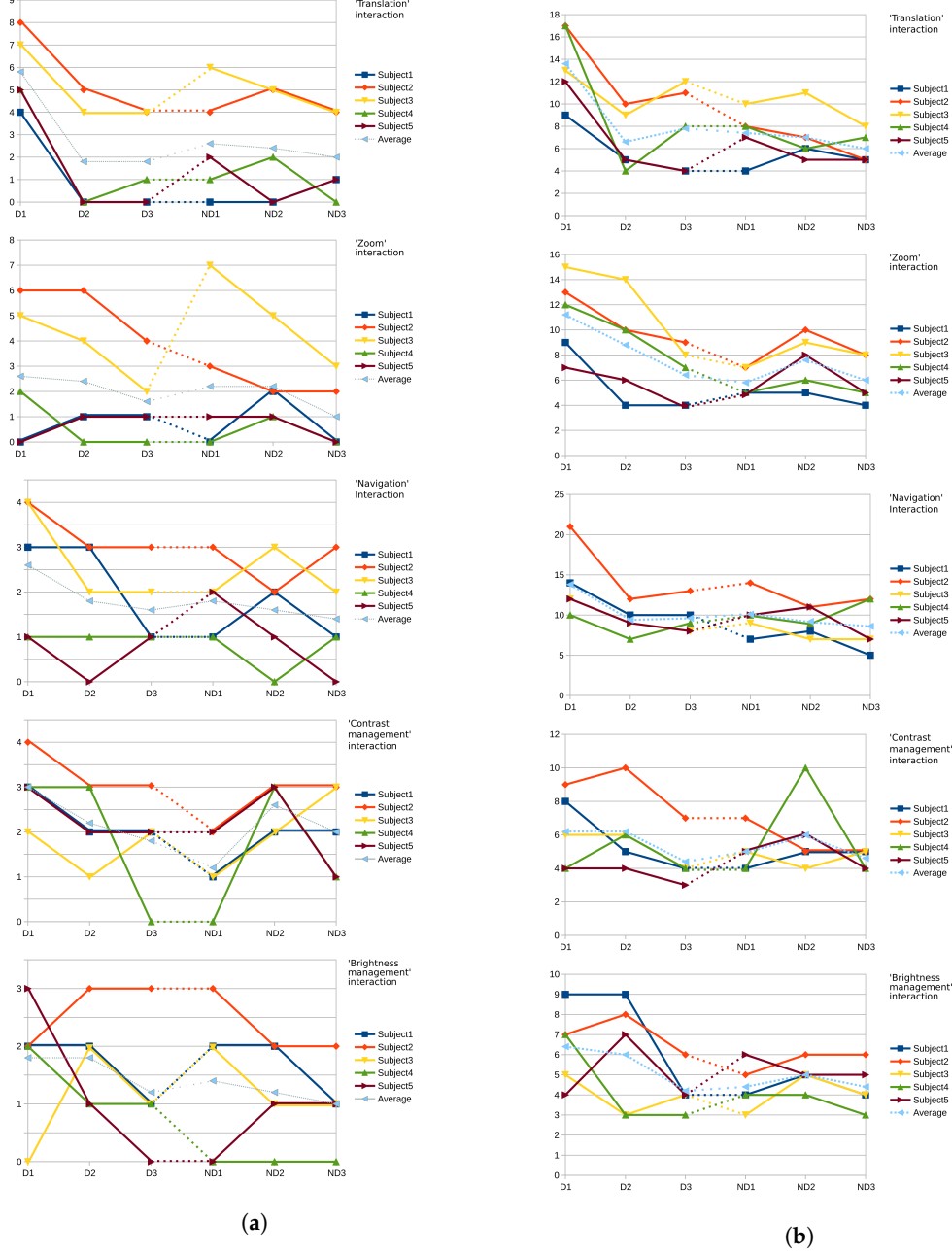

(**a**)

(**b**)

**Figure 3.** Evolution of "expert user" performance in terms of: "gaps" (**left**); and time of completion (**right**). (**a**) Number of 'gaps' per user, for each task type; (**b**) Time of completion (in seconds) per user, for each task type.

From the questionnaires presented to the subjects, we received the following feedback:

- **Translation** was deemed mostly comfortable and usable by the subjects.

- **Zoom** was deemed mostly comfortable while giving a satisfactory control over the task.
- **Stack Navigation** was deemed as offering a good control by the subjects.
- **Brightness/Contrast Management** were deemed to be well controllable and usable.

From this feedback, it appears that the refined vocabulary we proposed was well received comparably to the spontaneously propose gestures given during the "Wizard of Oz" phase.

### 6.3. Technical Limitations of the LeapMotion$^{TM}$

During both of our experiments, the subjects occasionally experienced hand detection difficulties that were inconsistently met throughout the whole proceedings. Some of these problems could be attributed to imperfections in the LeapMotion$^{TM}$ device's detection system, notably when switching finger postures, although one can decrease significantly these errors by learning to adopt more pronounced hand postures. However, it appeared that some of these detection issues were caused by natural lighting conditions (time of the day, weather, etc.) and whether the subject was wearing reflective accessories such as rings or reflective bracelet or watches. Even though the parameters can be controlled in the OR, they remained a constraint to be wary of. On a similar matter, the possibility of unintentional human interference made by another operating team member should be kept in mind and stresses the need for a dedicated (small) space of interaction. Of course, the LeapMotion$^{TM}$ offers a mono-user interaction by design, which was suitable for our work but also constraining the scope of this work to a single user context. Therefore, even though our approach and results are not device-bound, another detection system could be used to extend this work to develop approach that allows the design vocabularies in regard to a simultaneous multi-user context.

### 6.4. Comparison and Discussion Regarding a Similar Study

As mentioned above in this paper (Section 4), Jurewicz et al. [14] presented a similar work, constructed around a similar approach. It is important to note however that, for each study, even though they share the common domain of application, i.e. the operating room, the case of use is radically different. While our work focused on medical images manipulation, Jurewicz et al. focused on computer-assisted anesthetic tasks. Nonetheless, both studies aim to propose a relevant method to design gesture vocabularies for touchless interaction in surgical environment.

In both Jurewicz et al.'s study and ours, the principle relies on conducting a "Wizard of Oz" experiment on two populations of subjects: experts and non-experts. From our understanding, Jurewicz et al. considered both populations to be target users, using different vocabularies for each if the respective collected gesture vocabularies do not show enough similarities to converge. Our study, however, did not regard our "all-coming" (or non-expert) users to be final users. One of the key principles in our approach was that we, as explained, used the "all-coming" extracted gesture vocabulary to outline the base of an intuitive vocabulary that would be refined using the later feedback and comparison from our target users. We deemed this methodological divergence to be rooted in two different stances. Firstly, we would argue that, with most modern software interfaces, either the novice interface's set of available interactions is a subset of the interactions available to expert users, or both sets, if different, remain available at the same time. In any case, the richer and more numerous the functionalities are, the higher is the probability to stumble upon the problem of overlapping gestures for different interactions. Secondly, we believe that, in the highly specialized context that is the operating room, novice users either are users for a brief period of time, or in training and ultimately aim to become expert users. Another notable difference in our study is that, as mentioned in Section 5.1.3, we do not consider intuitiveness to be the only important criteria for the design of a relevant gesture vocabulary. Hence, the occurrence of our pre-defined gesture vocabulary, which is a component in our methodology that does not meet, we believe, any direct equivalence in the Jurewicz et al. study.

From our point of view, our study benefited from a case of use (medical images manipulation) that was less complex, in terms of number of interactions to map our vocabulary with. Images manipulation

is also composed of interactions that are admittedly more widespread than the interactions that have to be fulfilled to complete computer assisted anesthetics tasks. This likely facilitated our deduction of a gesture vocabulary based on novice users feedback and that closely matches the vocabulary that would appear fitting for final users. Jurewicz et al., on the other hand, met an increased number of interaction proposals that were significantly different between the ones proposed by the novice or expert users. This led us to consider the possibility that, using our approach for their case of use, we might have needed an extra step of "Wizard of Oz"-based protocol to properly refine our final user gesture vocabulary. It is also worth noting that Jurewicz et al. introduced metrics, such as, notably, the "reaction time" metric, as a measure of intuitiveness that could also have been useful to our work.

Nevertheless, the fact that both of our approach were met with success, while both being based on "Wizard of Oz", comfort us in our belief that this type of methodology is relevant regarding the design of effective and intuitive gesture vocabularies for touchless interaction in the operating room.

## 7. Ongoing Work and Perspectives

In this paper, we have presented our methodology to extract intuitive gesture vocabularies to implement effective and intuitive touchless interactions for a specific case of use. We carried out and implemented this methodology over one of our targeted cases of use consisting of the consultation of two-dimensional medical images of tomographic scans while working with the LeapMotion$^{TM}$. To this effect, we adapted our experimental protocol to this specific case of use and presented it to a group of 12 subjects. This allowed us to extract and isolate key gesture components on which to capitalize to deduce an intuitive gesture vocabulary that would be adapted regarding our selection of criteria, in the second experimental stage to the specific targeted context of use.

To validate our deduced gestures from the previous phase of our experiment, we presented it to a panel of five expert users (i.e., fellow surgeons). The feedback collected was perceived as being comfortable, and usable while offering a satisfying degree of control over the presented tasks. In addition, the time of completion for these tasks decreased, showing a positive learning effect. In addition, to back up this observation, we noted a general decrease in the number of errors (i.e., the number of "gaps") as the user's familiarity with the vocabulary grows.

To complete our goal, which was to propose an intuitive and efficient touchless interaction to be used in per-op by surgeons, we will have to apply our methodology to our second case of use that consists the consultation of 3D images. Once this is done, we will have to validate our extracted intuitive gesture vocabularies, as we did for the 2D case of use, by conducting usability tests on application prototypes integrating these vocabularies as interactive means involving the targeted users that are surgeons and staff of surgery teams.

**Author Contributions:** This work has been done as part of the main author A.H.'s PhD Thesis under the direction and supervision of co-author D.B.

**Funding:** This works is part of BPI 3D-Surg 2015-2019 project. Work-package no. 3.4.1- "New touchless interaction models" and work-package no. 3.4.2- "Prototyping and optimization for real-time interaction".

**Conflicts of Interest:** The authors declare no conflict of interest.

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
