# Peer review of "Approach for Intuitive and Touchless Interaction in the Operating Room"

_2571-8800, doi:10.3390/j2010005_

Round 1

Reviewer 1 Report

This study sought to introduce the conceptual development of a novel touchless technology to be used in sterile environment, such as the operating room (OR), in order to allow interaction without loss of sterility. The authors presented an intuitive “vocabulary” at the basis of the gesture recognition used in the very specific context of the OR. The authors reported many details about the proposed concept, including the methodological approach (User Centered Design and User Experience) used for the definition of the gesture vocabulary, the definition of specific criteria and the specific use case for 2D images. The authors then reported the use of a specific technology (i.e. Leap Motion) and a detailed evaluation protocol. Finally, the authors claimed the feedback collected was perceived as comfortable, usable while offering a satisfying degree of control over the presented tasks; they are working on an interface for 3D images.

General Comment

Although not extremely novel, this study addressed an interesting topic related to the need for touchless interaction in OR. In this case the authors proposed a methodological approach to design touchless system with basic gesture recognition to be used in sterile environment. From the technological point of view, the proposed solutions are not new at all, but it is interesting the overall description of the approach from a designer’s point of view. Although not standard, the structure of the article can be accepted as it is, since it is quite easy to follow. The English language is satisfactory.

Title

Ok. Just one suggestion. Since the journal is multi-disciplinary I would avoid very specific term such as “Wizard of Oz”, since the approach is not clear (although you explained it in the main text).

Abstract

Please try to introduce more information about the proposed approach and the main findings you obtained. Report also a one-sentence take-home message, as conclusion of your work.

Introduction

In general, ok. Introduction include “Context” and “Related works” sections. Please avoid informal writing.

Methods

Ok. Although the structure is not “standard” is quite easy to follow. The approach is clear as well as the assessment protocol

Results and Discussion

Ok. Please stress more the limitations of the use of Leap Motion technology in the setting of OR. Please compare your main findings with respect to the current literature. You should expand more the references.

Conclusions

Ok.

References

References should be extended.

Figures

Ok.

Tables

Ok.

Author Response

Response to Reviewer 1 Comments

Point 1:

Title

            Ok. Just one suggestion. Since the journal is multi-disciplinary I would avoid very specific term such as “Wizard of Oz”, since the approach is not clear (although you explained it in the main text).

Abstract

            Please try to introduce more information about the proposed approach and the main findings you obtained. Report also a one-sentence take-home message, as conclusion of your work.

Introduction

            In general, ok. Introduction include “Context” and “Related works” sections. Please avoid informal writing.

Methods

            Ok. Although the structure is not “standard” is quite easy to follow. The approach is clear as well as the assessment protocol.

Response 1: We have taken into consideration these remarks and subsequently modified the article’s title, and slightly tweaked our abstract. The article went under additional reviews by english speaking native peer, and we trust that the ensuing editing improved the overall quality of the writing.

Point 2: Please stress more the limitations of the use of Leap Motion technology in the setting of OR.

Response 2: It is true that the LeapMotion comes with a range of limitations that are worth mentioning. Therefore, we added a ‘Technical limitations of the LeapMotion’  subsection to cover this aspect.

Point 3: Please compare your main findings with respect to the current literature. You should expand more the references

Response 3: We enriched this version of the paper with a few references, notably a references you’ll find designated as “O’Hara et al. [8]” that is an imporant references that should have been indeed included from the start; we apologize for this oversight. Additionally we also added reference to a recent survey, designated as “Vogiatzidakis et al. [13]” published in the Multimodal Technologies and Interaction journal that led us to being able to compare our study with a similar one designated as “Jurewicz et al. [14]” that relied on a similar methodology in a very similar context.

Reviewer 2 Report

Summary:
In the paper "Wizard of Oz based approach for intuitive touchless interaction in the Operating room", an
approach is presented, which shall support surgeons in working with computer-based software applications
in a new way. The latter means that they do not have to interrupt their normal working procedure, this
is achieved by operating with the software applications in an intuitive, touchless manner. The authors well
present the current limitations and drawbacks when using such software applications in an operating room. Then,
their approach is described, accompanied by an experiment, which shows the applicability of the approach.

Points in favor:
    - Overall, the paper is written very well
    - The idea of the approach is making an contribution
    - Related work is discussed in a fine way
    - An experiment was conducted
    - The paper fits to the scope of the journal

Points against:
    - Some phrasings could be improved
       e.g., (1) the title, "Oz based" -> "Oz-based"
                                "Operating -> operating"
                                "intuitive touchless -> intuitive and touchless"
                                "Oz based is not clear in the title"
                 (2) "In during an intervention" in the abstract
         (3) "that is the Operating room" in the abstract
             ...
    - Please have a look at the commas in the entire work -> inconsistent usage
    - Please add a section at the end of the introduction that gives the structure for the paper
    - Contexts -> Introduction
    - I would summarize the related work within a table to better see the differences (just a suggestion)
    - Put some pictures of the experiment to the paper, gives a better impression

Author Response

Response to Reviewer 2 Comments

Point 1:

 - Some phrasings could be improved

e.g., (1) the title, "Oz based" -> "Oz-based"

"Operating -> operating"

"intuitive touchless -> intuitive and touchless"

"Oz based is not clear in the title"

(2) "In during an intervention" in the abstract

(3) "that is the Operating room" in the abstract

...

- Please have a look at the commas in the entire work -> inconsistent usage

- Please add a section at the end of the introduction that gives the structure for the paper

- Contexts -> Introduction

Response 1: We did present our article for language and phrasing review to english native peers. We believed the adjustments we consequently made, improved notably the overall quality of the writing. Also, a subsection has been added, in the newly renamed ‘Introduction’ section, that gives the structure of the paper.

Point 2:

- I would summarize the related work within a table to better see the differences (just a suggestion)

- Put some pictures of the experiment to the paper, gives a better impression

Response 2:

The nature of work and therefore results, revolving around methodological design, is rather more qualitative than quantitative. For the actual data, we have yet to find any work that uses the same criteria as we do. Comparing on these aspects would be difficult. I personally am refractory to the use of table to compare qualitative data.

Unfortunately, our experiment environment are, at the time, either not easily accessible (setup near the surgeons) either repurposed (setup concerning the «all-coming» subject). We, authors, didn’t reach a consensus whether or not we should re-create look alike pictures of the experiment. The pictures we’re holding currently are either low quality (coming from web-camera video footage) or not relevant in regard to the global scope of our work (thus anonymity regarding the publication of these pictures remains an aspect we’re unsure of).
